# IRE1a-Induced FilaminA Phosphorylation Enhances Migration of Mesenchymal Stem Cells Derived from Multiple Myeloma Patients

**DOI:** 10.3390/cells12151935

**Published:** 2023-07-26

**Authors:** Francesco Da Ros, Kinga Kowal, Carla Vicinanza, Elisabetta Lombardi, Francesco Agostini, Rosanna Ciancia, Maurizio Rupolo, Cristina Durante, Mariagrazia Michieli, Mario Mazzucato

**Affiliations:** 1Stem Cell Unit, Department of Research and Advance Cancer Diagnostic, CRO Aviano, National Cancer Institute, IRCCS, Aviano, Italy; francesco.daros@cro.it (F.D.R.);; 2Department of Life Sciences, University of Trieste, 34151 Trieste, Italy; 3Oncohematology and Cell Therapy Unit, Department of Medical Oncology, CRO Aviano, National Cancer Institute, IRCCS, Aviano, Italy

**Keywords:** multiple myeloma, mesenchymal stem cells, IRE1a, migration

## Abstract

Multiple myeloma (MM) is an aggressive malignancy that shapes, during its progression, a pro-tumor microenvironment characterized by altered protein secretion and the gene expression of mesenchymal stem cells (MSCs). In turn, MSCs from MM patients can exert an high pro-tumor activity and play a strong immunosuppressive role. Here, we show, for the first time, greater cell mobility paralleled by the activation of FilaminA (FLNA) in MM-derived MSCs, when compared to healthy donor (HD)-derived MSCs. Moreover, we suggest the possible involvement of the IRE1a-FLNA axis in the control of the MSC migration process. In this way, IRE1a can be considered as a good target candidate for MM therapy, considering its pro-survival, pro-osteoclast and chemoresistance role in the MM microenvironment. Our results suggest that IRE1a downregulation could also interfere with the response of MSCs to MM stimuli, possibly preventing cell–cell adhesion-mediated drug resistance. In addition, further investigations harnessing IRE1a-FLNA interaction could improve the homing efficiency of MSC as cell product for advanced therapy applications.

## 1. Introduction

Multiple myeloma (MM) is a hematological malignancy characterized by the abnormal proliferation of plasma cell clones and by a tropism for bone marrow tissue. The tumor is derived from asymptomatic conditions such as monoclonal gammopathy of uncertain significance (MGUS) and smoldering myeloma. In turn, it develops and modulates complex interactions with the bone marrow microenvironment that sustain its progression and drug resistance: this makes MM a treatable but not curable malignancy [1]. The microenvironment can be divided into a non-cellular component, composed by extracellular matrix protein with a rich fluid milieu, as well as by the cellular counterpart that comprises, among others, the bone marrow mesenchymal stem cells (BMMSCs) [2].

Evidence showed that BMMSCs and MM cells reciprocally affect each other to favor a pro-tumor environment [3,4,5]. The cross-talk between MM and BMMSCs can both be direct [6] and indirect, i.e., through growth factors and exosomes release [4,5]. This leads to chronic alterations in BMMSC physiology that induce a pro-inflammatory and immune-permissive environment [7,8]. Indeed, transcriptome analysis showed that the expression profile of MM-BMMSCs is markedly altered in comparison to healthy donor (HD)-derived BMMSCs [9,10], with particular regard to cell cycle progression, immune response and osteoblastogenesis. Such findings are in agreement with previously published papers showing the deregulation of osteoblast differentiation and increased IL-6 secretion in MM-BMMSCs [11,12,13,14].

As modifications imposed by MM cells [15] to the bone marrow microenvironment also confer chemoresistance to MM itself, it is very important to identify new anti-tumoral targets interfering with the altered MM microenvironment. In this context, the IRE1a-XBP1s axis has proven to be promising. Inositol-requiring enzyme 1 alpha (IRE1a) is an endoplasmic reticulum (ER) receptor involved in ER stress and in unfolded protein response (UPR) [16]. The activation of IRE1a occurs normally in UPR, together with the induction of the ATF6 and PERK pathway. In this context, IRE1a exerts different activities such as the induction of the regulated IRE1a-Dependent Decay (RIDD) of mRNAs, increase in XBP1s level through unconventional splicing and initiation of JNK cascade [16]. These events lead to the resolution of ER stress, reached with a reduction in mRNA translation and protein folding, as well as to cell survival. It is not surprising that this pathway, in particular the IRE1a-XBP1s axis, is activated in MM cells to avoid apoptosis caused by ER stress, due to abnormal IgG production, and that it is involved in MM chemoresistance [17,18,19,20]. Previous publications showed that the inhibition or disruption of the IRE1a-XBP1s axis in MM cells can reduce tumor progression [17,18,19] and can synergize with MM-specific drugs such as bortezomib and lenalidomide [17,20]. Interestingly, the activation of the IRE1a-XBP1s pathway in non-tumor cells of the MM bone marrow microenvironment can contribute to tumor progression. As an example, MM-secreted extracellular vesicles induce pro-osteoclastogenesis differentiation in precursor cells through the activation of the IRE1a-XBP1s axis [21]. Moreover, Xu and colleagues showed that a higher expression of XBP1s in MM-BMMSCs, compared to HD-BMMSCs, was associated with increased secretion of pro-tumor and of pro-osteoclast factors [22].

Even though there is a general consensus regarding differences between MM- and HD-BMMSCs, the impact of MM on patent’s BMMSC migration potential was poorly investigated. Considering the innate capability of these cells to home and reach cancer/wound sites [23,24] potentially exerting a direct pro-tumor effect [6], we aimed to explore the impact of the MM microenvironment on the MSCs’ migration properties. Interestingly, a recent paper shed light on a non-canonical role of IRE1a in the regulation of actin remodeling and migration process in vitro and in vivo. Indeed, Urra and colleagues discovered a non-canonical role of IRE1a in the control of migration in tumor cell lines through FilaminA (FLNA) phosphorylation [25]. FLNA mediates the adhesion of the cytoskeleton to the plasma membrane and interacts with several partners controlling actin remodeling and migration in both healthy and tumor cells [26,27,28,29]. The activity of FLNA is also controlled by several interacting proteins and by post-translational modifications [27]. In particular, the phosphorylation of Ser 2152 (S2152) is necessary to trigger an efficient migration process [25,30]. A mutated form of FLNA, unable to be phosphorylated at S2152 (S2152A), failed to rescue the migration impairment in vitro and in vivo [25] and reduced cell response to chemoattraction [30].

Thus, here we investigated for the first time whether the IRE1a-FLNA axis is involved in BMMSC migration and whether this pathway is modified by the MM microenvironment.

## 2. Results and Discussion

### 2.1. MM-BMMSCs Exert High Migratory Capability

MM is a complex malignancy relying on a dense network of interactions between tumor cells, microenvironment and BMMSCs. MSCs are pivotal components of the bone marrow microenvironment in both physiological and pathological conditions. For the first time, we investigated the impact of the MM microenvironment on BMMSC migration potential, a key biological property of such stem cells [23,24]. We found that MM-BMMSCs are characterized by enhanced mobility, when compared to HD-BMMSCs. In particular, 24 h after the application of the chemoattractive stimulus, MM-BMMSCs showed a two-fold increased migration potential vs. HD-BMMSCs (Figure 1A,B). Noteworthy, when comparing BM-MSCs isolated from 11 MM patients with HD-BMMSCs, 73% (i.e., eight out of eleven) showed a strongly increased migration potential while 27% (i.e., three out of eleven) displayed unaltered mobility (Figure 1C). We then attempted to relate MM-BMMSC migration potential to clinical aspects of our patients, but the analysis did not reveal a significant correlation between motility and MM stages (Spearman’s correlation r_s_ = 0, *p* = 1) or with applied chemotherapy (Spearman’s correlation r_s_ = 0.255, *p* = 0.45) (Table 1). Thus, despite the reduced sample size, we can suggest that a higher migration potential could be a common feature of MM-BMMSCs, independently of the patients’ clinical conditions. 

### 2.2. FLNA Is Phosphorylated in MM-BMMSCs

To further explore MM-BMMSC migratory behavior, we analyzed FLNA expression and phosphorylation levels. Indeed, FLNA mediates the adhesion of the cytoskeleton to the plasma membrane and its activity is mainly stimulated by phosphorylation [25,30]. Surprisingly, in the MM-BMMSC samples, an average upregulation of phosphorylated FLNA (Ser 2152) with a 1.9-fold increase when compared to HD-BMMSCs (Figure 2A,B) was observed. Despite FLNA activation, the level of total FLNA and PKCα, one of the kinases responsible for its phosphorylation [26], remained unaltered in MM-BMMSCs (Figure 2A,B).

Thus, when compared to HD-BMMSCs, we found a higher motility of MM-BMMSCs that, in turn, was associated with the upregulation of the phosphorylated fraction of FLNA. In agreement with other previously reported results [10,11,13], such MM-BMMSC behavior can be considered as cell-inherent, as it is maintained also ex vivo, i.e., far from the tumor site. We can speculate that such effect may depend on stable epigenetic modifications induced by MM cells [12,31]. Furthermore, as we observed that increased MM-BMMSC migration does not correlate with MM stages or therapy, we can hypothesize that related changes in FLNA phosphorylation could be an early modification occurring during pathology onset. In our opinion, such finding is particularly interesting, as it involves the cross-talk between MM cells and BMMSCs with potential implications on tumor growth progression and chemoresistance [5,32]. Due to their increased migration potential, a higher fraction of MM-BMMSCs could reach tumor cells, potentially promoting cancer growth by direct cell–cell contact or through a bystander effect [4,5,6]. Indeed, an enhanced response of BMMSCs to MM stimuli could be a component of the environment-mediated drug resistance which incorporates soluble factor-mediated drug resistance and cell adhesion-mediated drug resistance [33].

### 2.3. IRE1a-FLNA Interaction in Non-Migrating and Migrating BMMSCs

Recently, Urra and colleagues unveiled a new non-canonical role of IRE1a in the control of FLNA activation in tumor cell lines [25], which was not concerned with IRE1a kinase and RNAse activity. They showed that IRE1a is necessary for FLNA-mediated mobility changes in different tumor cell lines and in animal cells. Considering the activation of FLNA observed in MM-BMMSCs (Figure 2A,B), we investigated (i) whether IRE1a interacts with FLNA also in primary BMMSCs and (ii) whether this can play a role in the migration process. Performing immunoprecipitation assays (Appendix A), we found that IRE1a and FLNA interact in primary BMMSCs, and we showed that their binding increases during migration. In particular, we firstly observed that, in non-migrating conditions, IRE1a bound low amounts of FLNA in both HD- and MM-BMMSCs (Figure 3A). Still in non-migrating conditions, we observed variable levels of FLNA/IRE1a ratio in MM-BMMSCs without significant difference with HD-BMMSCs (Appendix A). In static conditions, in fact, the interaction between IRE1a and FLNA was generally reduced (Figure 3A and Appendix A), even though in BMMSCs derived from a few MM patients, a higher basal protein interaction was found. Otherwise, independently from baseline FLNA/IRE1a binding, we detected a striking increase (from 13- to over 100-fold) in FLNA/IRE1a interaction after migration toward chemo-attractive agents such as FBS and plasma bone marrow (PBM) (Figure 3B). Thus, we here report strong evidence showing that IRE1a is involved in the migration process of both primary human HD- and MM-BMMSCs, possibly functioning as a dock for FLNA activation. Our results are in agreement with evidence published by Urra and colleagues [25]. Noteworthy, in this present work, we took advantage of human primary cells and we analyzed only endogenous proteins, i.e., without any transient protein over-expression, to show the increased interaction between IRE1a and FLNA during the migration process (Figure 3A).

### 2.4. IRE1a Is Required for Efficient Migration of BMMSCs

We then investigated the potential impact of FLNA phosphorylation factors on the regulation of BMMSC migration. Surprisingly, comparing lysates from non-migrating and migrating BMMSCs, we detected no variation in S2152 phosphorylation and total level of FLNA (Figure 3C and Appendix A). On the contrary, migrating cells were characterized by significantly higher IRE1a level (Figure 3C) and a trend to statistical significance was observed (*p* = 0.06) for the PKCa level (Appendix A).

Thus, to challenge the hypothesis that IRE1a can be essential for the control of the migration process in BMMSCs and for the activation of FLNA, we downregulated IRE1a expression using a specific si-RNA directed against the *IRE1a* gene (Si-*IRE1a*). As expected, the reduction in IRE1a level, induced by Si-*IRE1a* (Figure 3D), decreased the migratory potential of BMMSCs in both HD and MM patients (Figure 3F and Appendix A). Interestingly, IRE1a downregulation lowered FLNA phosphorylation only during the migration process (Figure 3E) and not in non-migrating conditions (Figure 3D and Appendix A). Indeed, after Si-*IRE1a* transfection, the level of phosphorylated FLNA remained unaltered in non-migrating cells (Figure 3D) but it decreased when cells were subjected to a migratory stimulus: this supports the hypothesis that the FLNA/IRE1a axis can specifically control FLNA activation during the BMMSC migration process. In agreement with previous papers [25], IRE1a silencing did not completely blunt BMMSC migration, suggesting that other anaplerotic processes can contribute to the regulation of BMMSC migration.

### 2.5. Conclusions

In this work, we showed that the IRE1a-FLNA axis can play a pivotal, even though not exclusive, role in the control of BMMSC migration potential, which was in turn shown to be upregulated in MM-BMMSCs. Taken together, our results may have great importance in MM therapy. This work, in fact, can suggest the hypothesis to target IRE1a in order to counteract MM progression, and this is sustained by other published works reporting that the IRE1a-XBP1s axis is involved in MM progression and in patient outcome [34,35]. Moreover, the disruption of the IRE1a-XBP1s axis was shown to reduce the viability and drug resistance of MM cells in different contexts [17,18,19]. For example, anti-IRE1a drugs, such as nilotinib and KIRA8, showed beneficial effects against MM in synergy with bortezomib and lenalidomide [20]. On the other hand, it is known that IREa-XBP1s can regulate pro-tumor environment acting also in non-tumor cells such as osteoclasts and BMMSCs [21,22]. We are confident that IRE1a can be considered as a putatively common target to lower MM progression. We are aware that the reported results are part of a complex puzzle: to better understand the possible timing of IRE1a involvement in relation to MM, we will broaden our investigation also to BMMSCs derived from pre-MM conditions, such as MGUS and smoldering MM. In our opinion, this work can suggest that tIRE1a is an essential factor within MM pathophysiology.

Disentangling the IRE1a molecular pathway in HD- and MM-BMMSCs could also be helpful in other MM therapeutic approaches. In virtue of their cancer-specific homing potential, HD-BMMSCs can be applied as an advanced cell therapy product potentially counteracting MM progression through the modification of the cancer microenvironment. To achieve a therapeutically relevant medicinal product, ex vivo modifications of BMMSCs implementing their migration capacity are required [36]. Here, we showed that, in MM-BMMSCs, IRE1a can contribute to enhanced cell migration by binding active FLNA: such paradigm can be transferred to HD-BMMSCs. Priming such cells ex vivo in order to increase IRE1a-FLNA interaction could potentially improve the therapeutic activity of an advanced medicinal product based on HD-BMMSCs. Additional cell modifications conferring a clear citotoxic phenotype on MM target may also be required.

Active control of the IRE1a-FLNA axis can, thus, be considered as an important parameter in MM therapy, possibly requiring its direct downregulation in MM-BMSCs or its upregulation in HD-BMMSCs, when applied as an advanced medicinal product.

## 3. Material and Methods

### 3.1. Bone Marrow-Derived Mesenchymal Stem Cell Isolation and Culture

Human bone marrow aspirates were collected from patients with multiple myeloma with approval from the Ethics Board (CRO-2021-31). The samples were centrifuged and the upper fraction with the plasma bone marrow (PBM) was collected for further experiments. After the removal of the red blood cells with BD Pharm Lyse™ lysing solution (BD Biosciences, San Jose, CA, USA) and centrifugation at 1300 rpm for 10 min, the cell pellet was resuspended in a complete medium composed by Minimum Essential Medium α (α-MEM) supplemented with 10% fetal bovine serum (FBS) and 1% antibiotics (100 U/mL penicillin; 100 U/mL streptomycin) and plated on a culture dish. After 4–5 days, BMMSC colonies were visible. Adherent BMMSCs were expanded by standard procedures in complete medium.

### 3.2. Western Blot Analysis and Immunoprecipitation

To first compare BMMSCs from HD or MM patients, cells were seeded for three days and then were lysed with NP40 0.5% lysis buffer supplemented with metallo-protease and protease inhibitor cocktails (Merck KGaA, Darmstadt, Germany). The protein concentrations were measured using a BCA protein assay kit (Thermo Fisher Scientific, Waltham, MA, USA) and the proper amount was loaded into a 4–20% protean TGX gel (Bio-Rad, Hercules, CA, USA). After electrophoresis, the proteins were transferred to 0.2 μm PVDF membrane using a Trans-Blot Turbo device (BioRad, Hercules, CA, USA). The membrane was blocked with 5% milk (Blotting Grade Blocker -BioRad, Hercules, USA) in TBST and then the primary antibodies were added in 2.5% milk in TBST overnight at 4 °C. The used antibodies were anti-IRE1a (1:500, Cell Signaling, Danvers, MA, USA, #3294), anti-Filamin A (1:1000, Proteintech, Manchester, UK, #67133-Ig), anti-Phospho-Filamin A S2152 (1:1000, ABclonal, Woburn, MA, USA #AP0783), anti-PKC alpha (1:1000, ABclonal, Woburn, MA, USA #A11107) and anti-GAPDH (1:3000, Merkmillipore, Burlington, NJ, USA, #CB1001). After the incubation with proper secondary antibodies anti-mouse HRP (1:1000, GE Healthcare, Milan, Italy, #GENA931) or anti-rabbit HRP (1:1000, SouthernBiotech, Birmingham, AL, USA, #4030-05), the immunoreactive bands were visualized with Amersham ECL Prime Western Blotting Detection Reagent (Cytiva, Marlborough, MA, USA) and quantified using the ImageJ software.

For immunoprecipitation experiment, protein samples from migrating (after transwell migration) and not-migrating (plated in 24-well plates for the same period) conditions were incubated with 0.5 μg of anti-IRE1α antibodies (Cell Signaling, Danvers, MA, USA, #3294) in 500 μL of PBS and protease inhibitor cocktail for 3 h at room temperature with rotation. After this time, the protein–antibody complex was absorbed to 20 μL of Protein G sepharose 4 fast flow (Cytiva, Marlborough, MA, USA, 17-0618-01), and incubated for another 3 h at RT and subsequently overnight at 4 °C. Then, samples were centrifuged at 12,000× *g* for 20 s and the supernatant was collected for further analysis. After a wash with 500 μL of PBS and subsequent centrifugation, the pellet was resuspended in 20 μL of SB buffer 3×. Following this, Western blot analysis was performed as described above.

To investigate changes in protein expression induced by migration, we used protein samples from migrating (i.e., after transwell migration) and non-migrating cells (i.e., after seeding in transwell chambers in absence of chemotactic stimuli).

### 3.3. Migration Experiment

In all the experiments, 24-well transwell chambers with 8 μm pore membranes were used. 1.5 × 10^4^ cells were seeded in the upper chamber in 250 μL of α-MEM supplemented with 10% FBS. After 4–5 h, when all cells were attached to the membrane, the upper medium was changed with α-MEM without FBS. The migration stimulus was applied by adding the medium supplemented with FBS or with plasma obtained from MM patients in the lower chamber. After 24 h, the cells were fixed with 4% paraformaldehyde for 10 min and stained with 10% crystal violet for another 10 min. After two washes in PBS, images of total and migrated cells were acquired with Olympus CKX41 microscope (4× objective) equipped with Moticam S12 camera and Motic Images Plus 3.0 software. The migration potential was calculated comparing the area covered by migrated cells to the area covered by all the cells (migrated area/total area). For this analysis, we used ImageJ software. For the experiments with MM-BMMSCs, HD-BMMSCs were used as control.

### 3.4. Si-RNA Transfection

Anti-*IRE1a* si-RNA (Thermo Fisher Scientific, Waltham, MA, USA, ID S200432) was used to downregulate IRE1a in BMMSCs. Briefly, 2.5 × 10^4^ BMMSCs from MM patients and healthy donors were seeded in a 12-well plate. The next day, transfection was performed using Lipofectamine RNAiMAX (Thermo Fisher Scientific, Waltham, MA, USA, #13778) and Si-*IRE1a* according to the manufacturer’s protocol. After 72 h, the cells were collected and used for migration experiment and Western blot as described above. In all transfection experiments, BMMSCs (MM or HD) transfected only with Lipofectamine RNAiMAX were used as a control.

### 3.5. Statistical Analysis

For statistical analysis, unless differently described, a two-tailed *t*-test was performed. A value of *p* < 0.05 was considered statistically significant. Spearman’s correlation test was used to correlate migration with the clinical data reported in Table 1.

## Figures and Tables

**Figure 1 cells-12-01935-f001:**
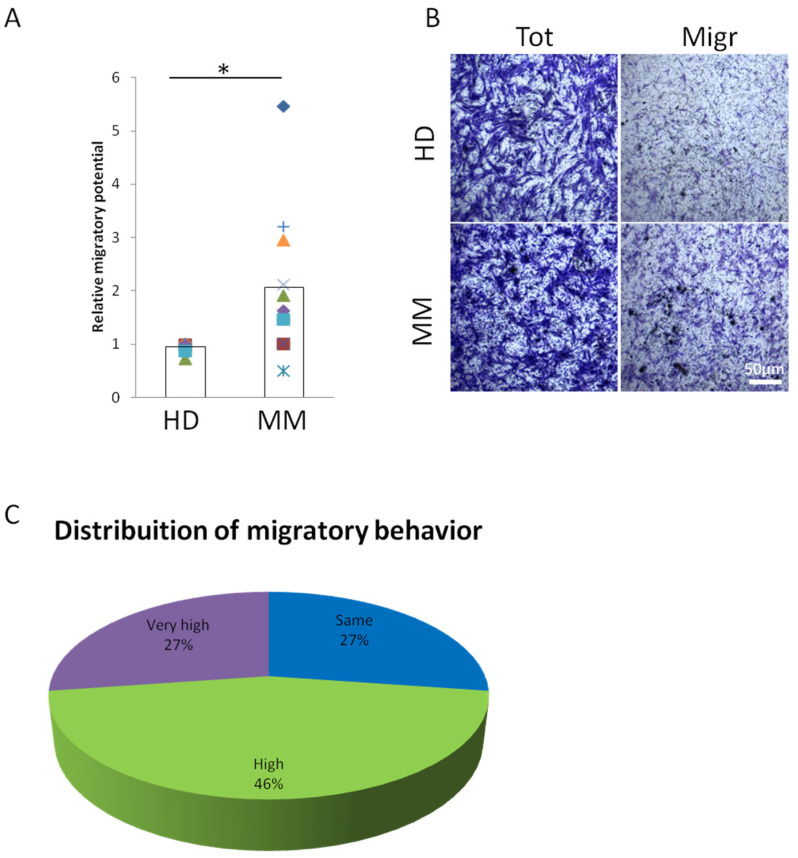
**Behavior of MM-BMMSCs during migration**. (**A**) Relative migration potential of MM-BMMSCs (*n* = 11) compared to HD-BMMSCs (*n* = 5) measured at 24 h by transwell migration assay. BMMSCs were seeded on transwell membranes with 8 μm pores; A-MEM supplemented with 10% fetal bovine serum (FBS) was used as chemo-attractive agent. Two-tailed *t*-test; * *p* < 0.05. (**B**) Representative images of total (left) and migrated (right) HD- and MM-BMMSCs after 24 h. Cells were stained by crystal violet. Scale bar = 50 μm. (**C**) Distribution of MM-BMMSCs (*n* = 11) described as having “Same” (i.e., from 0.5- to 1.5-fold), “High” (i.e., from 1.5- to 3-fold) or “Very high” (i.e., >3 fold) migratory behavior when compared to HD-BMMSCs.

**Figure 2 cells-12-01935-f002:**
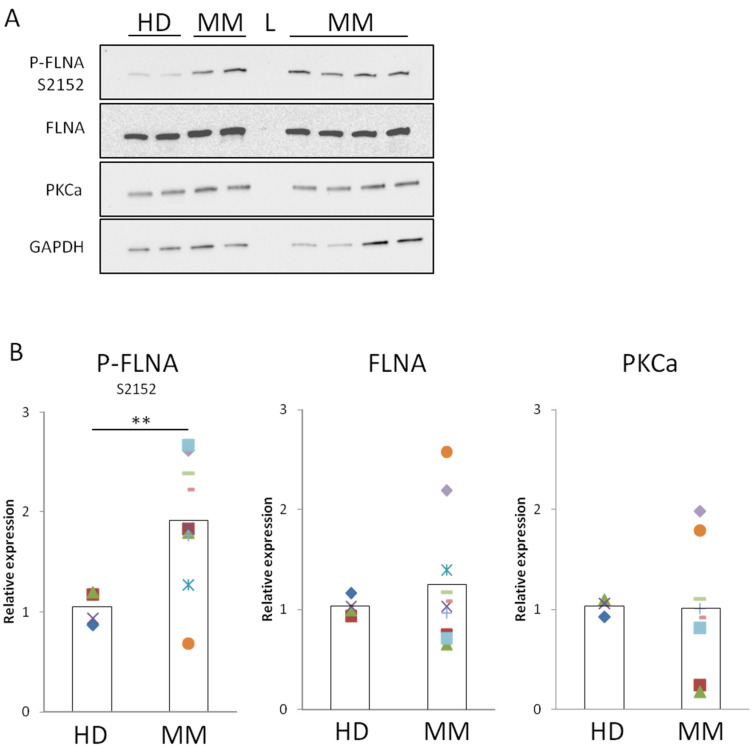
**MM-BMMSCs possess high level of active FLNA.** (**A**) Representative protein expression analysis by Western blot in lysates of HD- and MM-BMMSCs cultured for three days in static condition. GAPDH used as loading control. L = ladder. (**B**) Relative quantification of P-FLNA S2152, FLNA and PKCa expression in HD- and MM-BMMSC lysates. Images were analyzed with ImageJ software and P-FLNA S2152 level was normalized to FLNA signal, while FLNA and PKCa were normalized to GAPDH signal. Two-tailed *t*-test; **, *p* < 0.01. HD-BMMSCs *n* = 3/4, MM-BMMSCs *n* = 8/10, four experiments in triplicate.

**Figure 3 cells-12-01935-f003:**
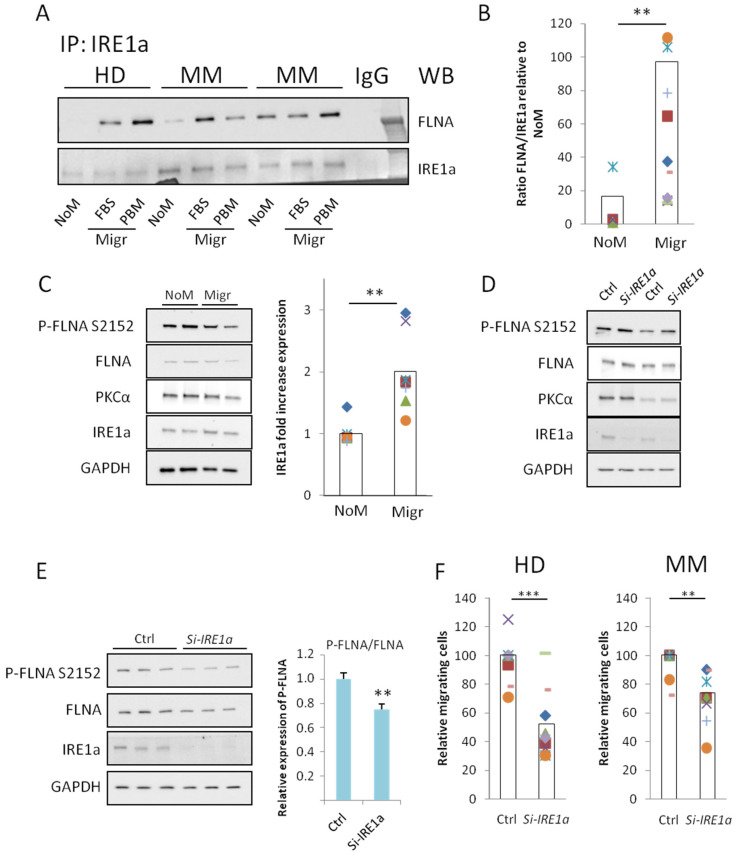
**IRE1a controls migration via FLNA bond and activation.** (**A**) Results of immunoprecipitation for IRE1a followed by WB analysis for FLNA and IRE1a itself in HD- and MM-BMMSCs both in non-migrating (NoM) and migrating conditions (Migr = FBS and PBM). After a 24 h transwell assay, BMMSCs were lysed and IRE1a immunoprecipitation was performed. IgG: ctrl with anti-IRE1a antibody without protein sample. (**B**) Relative quantification of the ratio between FLNA and IRE1a protein level in migrating (Migr = FBS and PBM) relative to non-migrating (NoM) condition. NoM, *n* = 5; Migr, *n* = 10. Two-tailed *t*-test; **, *p* < 0.01. (**C**) WB analysis of samples from BMMSCs in non-migrating (NoM) conditions and after 24 h of migration (Migr). A-MEM supplemented with 10% FBS was used as chemo-attractive agent. On the right, relative quantification of IRE1a level compared to NoM condition. *n* = 8. Two-tailed *t*-test, **, *p* < 0.01. (**D**) WB analysis in samples from BMMSCs after downregulation of IRE1a with Si-*IRE1a* for 72 h. Ctrl = only transfection agent. (**E**) WB analysis in BMMSCs after transfection with Si-*IRE1a* for 72 h and 24 h migration through 8 μm membrane; A-MEM supplemented with 10% FBS was used as chemo-attractive agent. On the right, relative quantification, with ImageJ, of P-FLNA level in Si-*IRE1a* samples compared to Ctrl. *n* = 9, one-tail *t*-test; **, *p* < 0.01. (**F**) Analysis of migration in HD- (left) or MM-BMMSCs (right) after downregulation of IRE1a with Si-*IRE1a*. BMMSCs were first transfected with Si-*IRE1a* for 72 h and then seeded on transwell membrane. A-MEM supplemented with 10% FBS was used as chemo-attractive agent. Relative migration to Ctrl cells was calculated. HD-BMMSCs: both Ctrl and Si-*IRE1a n* = 10; MM-BMMSCs: both Ctrl and Si-*IRE1a n* = 8. Two-tailed *t*-test; **, *p* < 0.01; ***, *p* < 0.001.

**Table 1 cells-12-01935-t001:** Migration potential of MM-BMMSCs and selected clinical features of each enrolled patient (R-ISS stage and therapy).

Patient	Migration	Stage (R-ISS)	Therapy
M1	++	2	Yes
M2	=	3	No
M3	+	Np before 2016	Yes
M4	=	3	Yes
M5	=	2	No
M6	+	3	No
M7	++	3	Yes
M8	+	2	Yes
M9	++	3a	No
M10	+	3	Yes
M11	+	3	Yes

=: same; +: high; ++: very high.

## Data Availability

Data is contained within this article and Appendix A.

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
