# Peer review of "IRE1a-Induced FilaminA Phosphorylation Enhances Migration of Mesenchymal Stem Cells Derived from Multiple Myeloma Patients"

_cells, 2023, doi:10.3390/cells12151935_

Round 1

Reviewer 1 Report

This manuscript entitled "IRE1a-induced phosphorylation enhances migration of MSC derived from multiple myeloma patients" by Francesco et al. showed that MSC from MM patients has higher migratory potential compared with healthy donors. Based on published literatures, the authors examined both whole and phosphorylated FLNA as well as the phosphorylation kinase PKCa level trying to connect the increased migratory potential with FLNA activation. Under migration state, the authors showed increased interaction between IRE1a with FLNA. Knockdown of IRE1a decreased the migration ability in both HD-MSC and MM-MSC. And they showed that knockdown of IRE1a also decreased the phosphorylation of FLNA under migration state but not non-migration state.   

However, the current data is not solid enough to support the title that IRE1a-induced phosphorylation enhances migration of MSC derived from MM patients. For example, does phosphorylation of FLNA is really required for the migration potential? Will phosphorylation-null mutant of FLNA will block the migration ability? How does IRE1a contribute to the phosphorylation of FLNA? 

Overall, the conception novelty of the manuscript is average. The writing of the manuscript was not rigor, and it’s a little hard to read as the writing with gramma errors. The Figure presentation was not rigor, for example lacking input loading control, negative control as well IgG control for IP. More work is required to make the data more solid to support their conclusions.

Majors:

1. Figure 1A, it is not clear what kind of test was performed to get the p value. Please define in the figure legend.

2. Figure 2B, it is not clear what kind of test was performed to get the p value. Please define in the figure legend.

3. The authors explore the S2152 phosphorylation of FLNA, it lacks background information, whether phosphorylation of FLNA is an indicator of activated form of FLNA and which phosphorylation sites have been reported to connect with the activation of FLNA activity. And mutation of S2152 that blocks the phosphorylation of S2152 will influence the migration of MM?

4. Figure 3A, The IP lack of Input as well as lack of IgG IP and negative control (for example, proteins that do not interact with IRE1a). It’s not clear when cell proceed to a migrative state, will the overall protein level of these examined protein altered? Also, did S2152 phosphorylation of FLNA increase under migration condition?

5. Figure 3C, 3D, using si-ERN1, different name to IRE1a, please clarify they are the same gene with different names in the main text.

6. Figure 3C, why the protein for two control lanes and two si-ENR1 lanes are quite different from each other. For examples, the PKCa protein level in the left two lanes was stronger than those in the right two lanes.

7. Figure 3D, for integrity of the data, the authors should also examine the total FLNA protein level as it is under migrative state. And it’s not clear how IRE1a contribute to the phosphorylation of FLNA.

8. The manuscript is drafted with lots of gramma errors. And it’s a little hard to read some sentences in the main text. Please edit the manuscript to make it more readable to the potential readers. 

Minors:

1. Lane 36, “in this contest”, did the author use the right word ‘contest’?

2. Lane 45, gramma error for this sentence. Please revise.

3. Lane 53, it should be “such as” not “as” to give the examples of anti-myeloma drugs. 

4. Lane 81, gramma error, it’s better to express as “could also influence” than “could influence also”.

5. Lane124, gramma error, “explore at molecular level the expression and activation of FLNA”. It is better to express as “explore the expression and activation of FLNA at molecular level”.

6. Lane 126, “several partners” not “several partner”.

7. Lane 167-169, the sentence “endogenous IRE1a bound slightly FLNA in primary BMMSC …” is not proper written. Please revise it.

8. It’s much better to use “Relative protein level of p-FFLNA” for Figure 3D.

9. Lane 174, gramma error for “With this results…”.

10. Lane 178, Please index the results “…bond IRE1a-FLNA changes during the migration process” with corresponding figures.

Lots of gramma errors in the writing. And it's little hard to read some of the sentences. Extensive editing of English language required.

Author Response

  1. Figure 1A, it is not clear what kind of test was performed to get the p value. Please define in the figure legend.

We added the missing information in figure legends

  1. Figure 2B, it is not clear what kind of test was performed to get the p value. Please define in the figure legend.

We added the missing information in figure legends.

  1. The authors explore the S2152 phosphorylation of FLNA, it lacks background information, whether phosphorylation of FLNA is an indicator of activated form of FLNA and which phosphorylation sites have been reported to connect with the activation of FLNA activity. And mutation of S2152 that blocks the phosphorylation of S2152 will influence the migration of MM?

We thank the reviewer for the comment. We added two sentences in the introduction and result section to explain the role of FLNA  and P-FLNA (S2152) in migration process. Moreover we mentioned two papers that describe how the mutation of S2152 site (S2152A) impairs the migration process in different cell lines and in vivo.

  1. Figure 3A, The IP lack of Input as well as lack of IgG IP and negative control (for example, proteins that do not interact with IRE1a). It’s not clear when cell proceed to a migrative state, will the overall protein level of these examined protein altered? Also, did S2152 phosphorylation of FLNA increase under migration condition?

Considering IP, in the Supplementary Figure 1A there was the image with IgG and input control. However to address the point and to improve the quality of the image, we changed Suppl Figure 1A with new IP for IRE1a, showing inputs, IgG and negative control. Moreover we changed the figure 3A, relative to IP in non migrating and migrating conditions, in order to show also the IgG control of IP.

Considering the modification of the overall protein level during migration, we thank the reviewer for the interesting question. To answer, we performed new WB experiments comparing the expression of IRE1a, PKCa, FLNA and P-FLNA in non-migration and migration conditions. These data are reported in the new figure 3C and suppl. Figure 1C. Even if we did not detected any modification in S2125 phosphorylation of FLNA and total FLNA level, we observed an increase of IRE1a expression during migration in all samples analyzed. This point is of particular interest because it suggests that IRE1a is crucial during the migration process to maintain, partially, the phosphorylation of FLNA. Indeed, as reported in Figure 3 E and 3 F, the downregulation of IRE1a induces a reduction of P-FLNA and an impairment in the migration response. In the main text we added this new result.

  1. Figure 3C, 3D, using si-ERN1, different name to IRE1a, please clarify they are the same gene with different names in the main text.

We considered the comment of the reviewer and, to avoid misunderstandings for the reader, we directly substituted ERN1 with the alias IRE1a in the whole text. 

  1. Figure 3C, why the protein for two control lanes and two si-ENR1 lanes are quite different from each other. For examples, the PKCa protein level in the left two lanes was stronger than those in the right two lanes.

We suggest that the difference in PKCa level (new figure 3D), and not of the other proteins, are due to variability of protein expression between different experiments. Indeed, each pair of CTRL/Si-IRE1a belongs to a different experiment and we noticed a small oscillation in PKCa comparing all experiments performed. For further images see the confidential image uploaded.

  1. Figure 3D, for integrity of the data, the authors should also examine the total FLNA protein level as it is under migrative state. And it’s not clear how IRE1a contribute to the phosphorylation of FLNA.

We added the total level of FLNA in the new Figure 3E (old figure 3D) as requested. Moreover we increased the number of experiments  (n from 3 to 9) for the new figure 3E and, for consitency with other figures, we analyzed the level of P-FLNA normalized for FLNA expression.

Considering the role of IRE1a in FLNA phosphorylation, Urra and colleagues showed data about its role and they suggested a possible mechanism that involves PKCa. Indeed, they demonstrated that not only the binding between IRE1a and FLNA, but also the dimerization of IRE1a is necessary for the phosphorylation of FLNA by PKCa. Thus, IRE1a acts as scaffold to allow proper FLNA phosphorylation during migration. Moreover, our results show that in migrating condition the cells possess high level of IRE1a/FLNA complex and they increase IRE1a expression to maintain the phosphorylation level of FLNA.    

  1. The manuscript is drafted with lots of gramma errors. And it’s a little hard to read some sentences in the main text. Please edit the manuscript to make it more readable to the potential readers. 

We performed an extensive editing of the whole text to correct errors and to make the paper more  readable. We submitted a clear version of the manuscript, with major modifications highlighted in yellow, and a revised version with all modifications visible.

Reviewer 2 Report

In this manuscript, Da Ros et al. have described the possible role of IRE1a-mediated migration of bone marrow-derived mesenchymal stem cells (BM-MSCs) of multiple myeloma (MM) patients proposing it as novel therapeutic targets and pathogenetic mechanisms. Contents are of very interest; however, their presentation should be improved by better introducing IRE1a and cytoskeleton proteins in MM in the introduction section. Moreover, figures shoul be better structured, such as Figure 1 by removing (C) and incorporating Table 1 transformed as a graph, or Figures 2 and 3 that might be combined. The link between some concepts is not that well-discussed, such as between the first and second section of the Results.

Please explain why in the original blot images no ladder is present for the molecular weights of proteins.

English should be improved and several typos are present

Author Response

In this manuscript, Da Ros et al. have described the possible role of IRE1a-mediated migration of bone marrow-derived mesenchymal stem cells (BM-MSCs) of multiple myeloma (MM) patients proposing it as novel therapeutic targets and pathogenetic mechanisms. Contents are of very interest; however, their presentation should be improved by better introducing IRE1a and cytoskeleton proteins in MM in the introduction section. Moreover, figures shoul be better structured, such as Figure 1 by removing (C) and incorporating Table 1 transformed as a graph, or Figures 2 and 3 that might be combined. The link between some concepts is not that well-discussed, such as between the first and second section of the Results.

We thank the reviewer for the suggestion. Considering Table 1 we decided to maintain in the main text, because it shows qualitative clinical data that would be hard to transform in a graph. We believe that a table can describe well the relationship between migration and clinical features of each patient. Moreover, we wrote the paper and created images to guide the reader step by step from the initial hypothesis to the final results. For this reason, we would prefer to maintain as separate figure 2 and figure 3 because they refer to different levels of the analysis. Moreover, with new experiments, Figure3 is richer of results and the addition of other graphs could be misleading for the reader.

We performed an extensive editing of the text, improving also the discussion of the results and the introduction session adding information about IRE1a and FLNA as suggested.

Please explain why in the original blot images no ladder is present for the molecular weights of proteins.

We uploaded the original blot images, containing also new experiments, with the missing information about the molecular weight.

Reviewer 3 Report

In the present manuscript, Francesco et al. investigated the role of IRE1a-FLNA axis in migration process of mesenchymal stem cells derived from multiple myeloma patients (MM-BMMSC).  MM-BMMSC have critical roles in cell growth, secretion of extracellular vesicles, and cell-adhesion mediated drug resistance (CAM-DR) of MM cells in BM microenvironment.  Although previous studies revealed that MM-BMMSC differed from healthy donor MSC in several aspects, the migration potential of MSC has not been analyzed.  In the present study, the authors showed that IRE1a-induced FLNA phosphorylation enhances migration of MM-BMMSC for the first time.  In conclusion, the authors suggest that anti-IRE1a approach could interfere with the response of MSC to MM stimuli, possibly impacting on CAM-DR. 

This reviewer thinks that the data presented include novel findings and the experiments are well designed and properly conducted, leading to reasonable and clinically relevant conclusions.  This reviewer has the following suggestions that may help to improve the manuscript.

1. The authors should show that anti-IRE1a activity could inhibit migration potential of MM-BMMSC by using FDA approved drugs including nilotinib, bortezomib, lenalidomide, and their combination.

2. The authors should show that nilotinib, bortezomib, lenalidomide, and their combination could decrease FLNA phosphorylation in MM-BMMSC.

Author Response

  1. The authors should show that anti-IRE1a activity could inhibit migration potential of MM-BMMSC by using FDA approved drugs including nilotinib, bortezomib, lenalidomide, and their combination.
  2. The authors should show that nilotinib, bortezomib, lenalidomide, and their combination could decrease FLNA phosphorylation in MM-BMMSC.

We thank the reviewer for the interesting suggestion and we partially answered to the question. Among the drugs suggested to inhibit IRE1a activity, only nilotinib can have a direct effect on IRE1a acting as tyrosin kinase inhibitor. Indeed bortezomib acts as proteasome inhibitor while lenalidomide has anti-angiogenic and immunomodulatory properties. However the mechanism that involves IRE1a in FLNA phosphorylation, does not require IRE1a phosphorylation but only its dimerization (Urra et al., 2018). In addiction the site for FLNA phosphorylation is a serine (S2152) and so we can exclude a possible effect of nilotinib in modulating this phosphorylation.

However, in order to address the point, we decided to use bortezomib to treat 3 lines of BMMSC and to control the overall protein expression and the migration properties. The choice of bortezomib was due also to the physical properties of the drug. Indeed bortezomib is liquid while nilotinib and linalidomide are in pills and it is very hard to obtain them and to prepare a liquid solution to use in vitro. After 24h of treatment with two different doses of bortezomib (20nM and 100nM), reported to reduce survival of MM cells (Obeng et al., 2006), we performed WB and migration analysis in BMMSC. After bortezomib treatment, we observed an induction of ER stress, proved by the augment of IRE1a. No variation in P-FLNA was detected. These results are in line with what reported In literature about the effect of proteasome inhibitor on ER stress induction (Obeng et al., 2006; Ri Masaki 2016). Considering the migration potential, we detected a reduction of BMMSC migration after bortezomib treatment, especially with the higher concentration. However, we believe that the observed reduction of mobility depends by the suffering condition induced by bortezomib, as show by crystal violet staining, and not by a direct effect on migration potential.

In the end, we think that these new results, even if interesting, don’t add new information for the focus of this paper but could, however, shed new light in the pursuance of the project. For this reason we submit these confidential data to the reviewer, but without adding them to the paper.

Urra, H, Henriquez, D R, Cánovas, J, Villarroel-campos, D, Carreras-sureda, A, Pulgar, E, Molina, E, Hazari, YM, Limia, CM, Alvarez-rojas, S, Figueroa, R, Vidal, RL, Rodriguez, DA, Rivera, CA, Court, F, Couve, A, & Qi, L. 2018. IRE1α governs cytoskeleton remodelling and cell migration through a direct interaction with filamin A. Nature Cell Biology, 20, 942-953.

Obeng, E. A., Carlson, L. M., Gutman, D. M., Harrington, W. J., Lee, K. P., & Boise, L. H. (2006). Proteasome inhibitors induce a terminal unfolded protein response in multiple myeloma cells. Blood, 107(12), 4907–4916. https://doi.org/10.1182/blood-2005-08-3531

Ri, M. (2016). Endoplasmic-reticulum stress pathway-associated mechanisms of action of proteasome inhibitors in multiple myeloma. International Journal of Hematology, 104(3), 273–280. https://doi.org/10.1007/s12185-016-2016-0

Round 2

Reviewer 1 Report

N/A

Reviewer 3 Report

-